# Inhibition of Fungal Strains Isolated from Cereal Grains via Vapor Phase of Essential Oils

**DOI:** 10.3390/molecules26051313

**Published:** 2021-03-01

**Authors:** Tereza Střelková, Bence Nemes, Anett Kovács, David Novotný, Matěj Božik, Pavel Klouček

**Affiliations:** 1Department of Food Science, Faculty of Agrobiology, Food and Natural Resources, Czech University of Life Sciences Prague, Kamýcká 129, 16500 Prague, Czech Republic; strelkova@af.czu.cz (T.S.); kovikussz@gmail.com (B.N.); kovacs.anett007@gmail.com (A.K.); bozik@af.czu.cz (M.B.); 2Department of Ecology and Diagnostics of Fungal Pathogens, Crop Research Institute, Drnovská 507/73, 16106 Prague, Czech Republic; novotny@vurv.cz

**Keywords:** antifungal, cereal, essential oil, fast screening, fungi, inhibition, vapour phase

## Abstract

Fungal contamination in stored food grains is a global concern and affects food economics and human and animal health. It is clear that there is a need to develop new technologies with improved performances that are also eco-friendly in nature. Due to the bioactivity of essential oils (EOs) in the vapor phase, their low toxicity for humans, and their biodegradability and antifungal properties, EOs could be a suitable solution. In this study, we explored the potential of thyme, oregano, lemongrass, clove, and cajeput EOs in the vapor phase. For 17 days, inhibitory activity was assessed against five strains of postharvest pathogens—*Aspergillus* spp., *Fusarium* s. l. spp., and *Penicillium*
*ochrochloron*—isolated from cereal grains. A modified disc volatilization method was used, which is more effective in comparison to traditional screening methods. Three concentrations were tested (250, 125, and 62.5 μL/L). The two highest concentrations resulted in complete inhibition of fungal growth; however, even 62.5 μL/L showed a significant antifungal effect. The efficiency of EOs followed this order: thyme > oregano > lemongrass > clove > cajeput. From our findings, it appears that the use of EOs vapors is a better option not only for laboratory experiments, but for subsequent practice.

## 1. Introduction

Food grains constitute a vital part of the daily diet of the population worldwide [1], and the most cultivated crop is wheat [2]. However, cereal grains contain a large number of microorganisms that deteriorate the products’ nutritive value and are dangerous to human and animal health [3]. Fungal contamination in stored food grains is a global concern and affects food economics both directly and indirectly [1]. Approximately 20% of wheat that would otherwise be available each year is lost due to diseases [2,4]. In addition, the mycotoxins secreted by different seed-borne fungi cause qualitative losses of commodities, and potentially induce various health problems in consumers [5]. Approximately 25–40% of cereals consumed all over the world are contaminated by mycotoxins [5,6], and the European Food Safety Authority, in their Panel on Contaminations in the Food Chain [7], stated grains and grain-based products are one of three main chronic dietary sources of ochratoxin A, a mycotoxin of *Aspergillus ochraceus*. Among the mycotoxins, aflatoxins chiefly produced by *Aspergillus flavus* are the most dangerous, and approximately 4.5 billion people in underdeveloped countries are exposed to aflatoxicoses [6].

However, the major factors responsible for fungal growth depend on various intrinsic and extrinsic factors, such as the meteorological conditions during vegetation and harvesting, the duration of storage, the water content of grains, the storage temperature, the humidity during storage, and the type of storage technology [1,3]. *Fusarium* spp., *Aspergillus* spp., and *Penicillium* spp. were found to be dominant [1,3,8,9], but the higher incidence of *Aspergilli* than other fungi may be due to their saprophytic nature and ability to colonize diverse substrates because of secretions of various hydrolytic enzymes [5]. Although *Fusarium* species are predominantly considered to be field fungi, it has been reported that the production of fumonisins (*Fusarium* mycotoxins) can occur post-harvest when storage conditions are inadequate [10]. The removal of mycotoxins from the food chain is one of the major challenges for food scientists. Therefore, efforts should be made to prevent sources of mycotoxins from being present, i.e., the mycotoxin-producing fungi in the stored grains [1].

Today, synthetic pesticides play a major role in crop protection, but the widespread use of pesticides has resulted in the development of pest resistance, outbreaks of new pests, toxicity to non-target organisms, and harmful effects on the environment [11]. During storage, phosphine (PH_3_) and methyl bromide (CH_3_Br) are used for grain disinfection, but limitations regarding the application of the latter in Europe are increasing, and in America, their use is totally banned [3]. Hence, there is a need to develop new fungicides/preservatives with improved performances that are also eco-friendly in nature [6].

Many studies have explored various nonchemical fungi management practices, including traditional methods such as drying to a safe moisture level, aeration, and dry heating and novel control measures such as hermetic storage, microwave heating, and applications of gaseous ozone, cold plasma, ionizing radiation, pulsed light, or supercritical carbon dioxide (SCeCO_2_) [1]. In addition, the use of plant extracts and essential oils (EOs) for the control of seed-associated fungi could be an eco-friendly solution, resulting in a lower chance of pathogens developing resistance [4,8].

EOs are volatile, oily liquids extracted from various plant materials; they are complex mixtures of chemical compounds with predominant terpenes associated with alcohols, aldehydes, and ketones [4]. Due to their bioactivity in the vapor phase, their low toxicity for humans, and their biodegradability and antifungal properties, EOs could find applications as fumigants for the protection of cereals and cereal-based products [12,13,14]. The antimicrobial effect could possibly be attributed to the presence of various antifungal substances, mainly phenolic compounds such as the monoterpenes thymol and carvacrol [15]. It is postulated that EOs, through their lipophilicity, have the ability to penetrate the plasma membrane, causing morphological changes in the hyphae, damaging the enzymatic cell systems, disrupting in the plasma membrane, and eventually destroying the mitochondria, thereby killing the fungi [1,8]. Many tested EOs show inhibitory effects on fungal postharvest pathogens, always in a dose-dependent manner [5,9,10,11,16,17].

The determination of the minimum inhibitory concentration (MIC) is important for setting a minimum dose for controlling fungal populations while using the lowest possible amount of pesticide [5]. The most common methodology for testing antifungal properties is the poisoned food technique—the use of culture media mixed with different amounts of an EO [2,18,19]. However, the less explored use of EOs in the vapor phase [13,20,21] seems to be a better option, as it should be more practical and more realistic for applications during storage. The main aims of the present study were to determine the antifungal activities of thyme, oregano, clove, lemongrass, and cajeput EOs in the vapor phase, and their suitability for use as disinfectants against *Aspergillus* spp., *Fusarium* spp., and *Penicillium ochrochloron* isolated from cereal grains.

## 2. Results

### 2.1. Compositions of Essential Oils

The major components of the EOs tested were identified and assessed by GC–MS. The main compounds found in thyme oil were thymol (58%), p-cymene (22%), and linalool (3%). Carvacrol was identified and determined as the major component of oregano oil (70%), followed by p-cymene (11%) and thymol (3%). Lemongrass oil contained, above all, geranial (42%) and neral (28%), and in smaller quantities geraniol (5%) and geranyl acetate (4%). The major components of clove oil were eugenol (80%), eugenol acetate (7%), and caryophyllene (7%). The minor components are listed in an earlier related publication [13]. The main components of cajeput oil were terpinen-4-ol (44%), γ-terpinene (20%), and p-cymene (14%); α-terpineol (4%) and 1,8-cineole (3%) were present in smaller quantities.

### 2.2. Antifungal Effect of the Essential Oils

After testing the highest concentration, namely, 250 µL/L, it was found that all of the EOs completely inhibited all strains. Therefore, 250 µL/L of any EO demonstrated fungicidal effect against the fungal strains we selected. Due to this finding, it was possible to start testing a lower concentration.

A lower concentration—125 µL/L—of all of the EOs except cajeput fully inhibited the mycelial growth of all strains. The cajeput EO was the least effective, so testing with a lower concentration of this EO was not appropriate.

Using the lowest concentration, namely, 62.5 µL/L, all of the EOs showed some inhibitory effect. A heat map of these results is available in Appendix A. The most effective was the thyme EO (Figure 1a), followed by the oregano EO (Figure 1b). These two EOs had the strongest inhibitory effect on the mycelial growth of every fungal strain. Among the strains tested, *A. niger* and *A. flavus* were the most resistant (significantly) and had almost the same mycelial growth development. On the sixth and seventh days, for the first time, oregano treatment on both *Aspergilli* did not statistically differ from the treatments of clove and lemongrass, respectively. Two days later, there was no statistical difference between the clove, lemongrass, oregano, or thyme treatments. However, the thyme EO as the first and the oregano EO as the second were still the most effective EOs. Control-equivalent full growth (4.75 cm diameter) for the thyme and oregano EOs was reached after 13 and 10 days of incubation, respectively. In the case of the thyme EO against *Fusarium sporotrichioides* and *F. solani*, full growth was not observed, even after 17 days. In general, the mycelia of the *Fusarium* strains grew very slowly due to the treatments of the thyme and oregano EOs. Until the tenth day, they were the most susceptible strains; after the tenth day, faster growth was observed. This mycelial growth development was significantly different compared to the controls (Figure 1e). However, no statistical difference from the control was observed at the end of the experiment (day 17). A table with statistical comparisons of the EOs’ efficiencies is available in the Appendix A.

The lemongrass EO showed a lower inhibitory effect, with similar activity against *F. sporotrichioides*, *F. solani*, and *A. niger* (Figure 1c). A very similar growth pattern was observed—intensive growth from the third day to the maximum on the sixth day—which means it delayed the fungal growth on average for two days compared to the control. The most resistant strain was *A. flavus*.

The clove EO (Figure 1d) at this concentration was the least effective EO against all of the tested strains, except *A. flavus*, against which the least effective was lemongrass EO. After 24 h, only the clove treatment was statistically equal to the growth control. Among the EOs tested, the clove EO was significantly the least effective against *F. solani*, whose mycelial growth was comparable to that of the controls (entirely grown on the fourth day). From the fifth day, the clove and lemongrass treatments did not significantly differ from the control for all fungi except *P. ochrochloron*. Of all the strains tested, *P. ochrochloron* was the least invasive and the most susceptible. The development of this fungus after treatment of EOs was significantly lower; even at the end of the assay, the mycelial growth did not reach the maximum. Additionally, the controls indicated that this strain is less invasive than others. Only when treated with the thyme or oregano EO was the mycelial growth of *P. ochrochloron* less inhibited than the growth of *F. sporotrichioides* and *F. solani*, but after the tenth day, in contrast to *Fusaria*, it did not increase pronouncedly. To briefly summarize the effectiveness of the EOs, they can be sorted in this order: thyme > oregano > lemongrass > clove.

## 3. Discussion

In grain cereals, mycotoxins are produced by fungal species such as *Aspergillus*, *Penicillium*, and *Fusarium* that colonize the plants in a field and can spread during the post-harvest period [22]. Spoilage of stored food commodities is a chronic problem and can produce qualitative and quantitative losses throughout the world [23]. Most EOs are considered to be “generally recognized as safe” (GRAS) food additives by the Food and Drug Administration (FDA), which makes them potential bio-resources of eco-friendly antifungal agents [24].

However, the dissimilar methods used for monitoring of antifungal efficiency constitute a problem. No standardized test has been developed and adopted for evaluating the possible antifungal activity of EOs against seed-borne fungi [25]. Not many studies have been performed that have applied EOs via gaseous contact, or have used similar EOs and similar strains as in this study. Until now, studies have been carried out using a combination of the same strains and different EOs or vice versa, and of varying concentrations. Herein, we used a method by Kloucek et al. [21]—a modified version of the commonly used disc volatilization method that uses a four-section Petri dish, a large filter paper disc evenly impregnated with EO, and a medium-containing lid. In comparison to the normal disc volatilization method, the labor and materials needed are reduced by several fold, and the composition of headspace is more uniform than in the case of a 6 mm disc, wherein different volatilities of particular compounds could influence the results. On the contrary, a number of studies have been performed using contact assays, such as the poisoned food technique. However, several researchers have concurred that the best antifungal activity of volatile compounds is achieved by gaseous contact, as opposed to aqueous solutions or agar contact [25,26,27,28,29,30].

Tullio et al. [27] reported that the inhibition effect of certain EOs (thyme red, clove, etc.) in the gaseous phase is generally higher than that in liquid state. They tested a concentration ranging from 10 mL/L to 19 µL/L against some fungi of the species *Mucor*, *Rhizopus*, *Penicillium*, *Alternaria*, and *Cladosporium*. In the disc volatilization method study of *Mentha piperita* EO, Tyagi and Malik [31] found that the minimum inhibitory concentrations (MICs) for *A. flavus* and *A. niger* varied from 1130 to 2250 µL/L, and the minimum fungicidal concentrations (MFCs) were 1130–2250 and 2250–4500 µL/L, respectively. In all strains, the zone of inhibition resulting from the exposure to EO vapors was again significantly larger than that due to same concentration of EO in the liquid phase, which supports our experiment. Yahyazadeh et al. [32] reported the fungicidal effect of thyme and clove EOs applied by gaseous contact against *Penicillium digitatum.* These EOs completely inhibited fungal growth by their volatiles at 340 µL/L, but when they were added to medium (direct contact), even at 600 µL/L, the effect was just fungistatic. However, even at concentrations that caused less than 100% mycelial growth inhibition, which is also our case, conidia lost their pigmentation (became hyaline). According to Yigit et al. [33], this effect might decrease the virulence of pathogens. In the case of clove EO, Bluma et al. [34] found that this oil did not show homogeneous antifungal activity against *Aspergillus Flavi*, and its efficacy depended on the water activity. In our study, evaluation of water activity was not included. However, the efficiency of the clove EO could have been affected due to this.

Additionally, a few contact assay studies and their results can be mentioned here. Linde et al. [35] used a modified microdilution technique and found that EO from *Petroselinum crispum* (parsley) exhibited fungistatic activity against all tested fungi, mainly *P. ochrochloron*. The in vitro results of Jahani et al. [36] showed that the growth of *A. niger* was completely inhibited by the contact assay with clove EO at concentrations of 200, 400, 600, and 800 µL/L on the first and tenth days, and thyme EO application at 800 µL/L on the tenth day. Morphological evaluation performed by both light microscopy and scanning electron microscopy conducted by Kohiyama et al. [37] showed that the antifungal activity of thyme EO against *A. flavus* could be detected at a concentration of 50 µL/L and the fungicidal effect at 250 µL/L. Oliveira et al. [24] also tested thyme EO, but at a higher concentration of 500 µL/L, and observed the complete inhibition of *A. flavus* growth. However, this is a very high concentration that cannot be used in practice. Krzyśko-Łupicka et al. [38] compared lemongrass, thyme, and cajeput EOs against *Fusarium* phytopathogens by a poisoned substrate assay and found out that thyme oil fully inhibited the growth at the lowest concentration (250 µL/L), lemongrass oil caused the same result at a slightly higher concentration (500 µL/L), and cajeput had a weaker effect even at 5 mL/L, which is consistent with its exclusion from our testing.

The determination of the antifungal activity of EOs appears to be influenced by the method used, as evidenced by the differences between the results just cited. It is also possible to assume that different chemical composition of specific tested EOs and different sensitivity or resistance of individual fungal strains are influential factors. Tullio et al. [27] described the content of thymol, a monoterpene phenol occurring in thyme EO, as a major contributor of bioactivity, and its connection with the efficacy of thyme EO, which corresponds to our findings. When multiple species of thyme EO were tested [21], a consistently significant inhibitory effect was obtained, although the composition differed. In addition, thymol was the most abundant compound in our EO. However, even EO rich in components such as carvacrol, p-cymene, or geranial has good preconditions for antifungal action. A eugenol-rich EO should act in the same way, in our case of clove EO; however, the factors mentioned above play important roles.

The advantages of using the volatile gas phase of EOs for agricultural and food products are that they may have less of an influence on the final taste and aroma, and their release may be regulated [34]. In addition, our results suggest that, thanks to the use of the gas phase, it is possible to achieve the inhibition of fungal growth using significantly lower concentrations than have been tested thus far. The above-mentioned studies have tested substantially higher concentrations by applying a gas phase or by direct contact, the efficiency of which could have been higher, but the need remains to find the lowest possible usable concentration. Higher concentrations could adversely affect sensory properties and would require more EOs to be used, which would be reflected in costs. In addition, our method makes it possible to monitor the inhibition of fungal growth over time, from which the need for possible repeated applications of EOs to different strains can be deduced. This could then be carried out more efficiently in practice, thanks to the use of the gas phase.

## 4. Materials and Methods

### 4.1. Essential Oils

Six essential oils were tested for their antifungal potential. The EOs from thyme (*Thymus vulgaris* L.) and lemongrass (*Cymbopogon citratus* L.) were purchased from the commercial supplier Sigma-Aldrich (Hamburg, Germany); oregano (*Origanum vulgare* L.), clove (*Syzygium aromaticum* L.), and cajeput (*Melaleuca alternifolia* Cheel) were purchased from Biomedica (Prague, Czech Rep.). These EOs were chosen according to previous results [21]. All of the used EOs were stored in glass bottles at 4 °C until used.

### 4.2. Essential Oil Analysis

Relative proportions of essential oils constituents were assessed by gas chromatography with flame ionization detector (GC–FID), Agilent 7890A (Agilent Technologies, Palo Alto, CA, USA) with HP-5MS, 30 m × 0.25 mm, and a 0.25 µm film thickness. The oven temperature was increased from 60 °C at a rate of 3 °C/min to a maximum of 231 °C, where it was kept constant for 10 min. Prior to the analysis, the EOs were diluted in hexane to a concentration of 1 µL/L. One microliter of the sample was injected in the split mode 1:12. The carrier gas was nitrogen (constant flow of 1 mL/min, 99.999% purity); the injector and detector temperatures were 250 °C. The relative proportions were calculated by dividing the individual peak area by the total area of all peaks; the response factor was not taken into account. Only compounds over 3% were included. The EO constituents were identified by mass spectrometry (GC–MS), the results of which were previously reported [13].

### 4.3. Microorganisms

The fungal strains used in present study were isolated in our laboratory from grain samples of wheat (*Triticum aestivum*), barley (*Hordeum vulgare*), and triticale (*Triticosecale*) of organic quality collected from experimental station of the Czech University of Life Sciences, Prague—Uhříněves. These included *Aspergillus flavus* Link (strain number VURV F-778), *Aspergillus niger* Tiegh. (strain number VURV F-779), *Penicillium ochrochloron* Biourge (strain number VURV F-780), *Fusarium sporotrichioides* Sherb. (strain number VURV F-804), and *Fusarium solani* (according to the new taxonomy—*Neocosmospora solani* (Mart.) L. Lombart and Crous). The fungal isolates were grown on Sabouraud Dextrose Agar (SDA) (Oxoid CZ s.r.o., Brno, Czech Rep.) at 25 °C. Pure cultures were obtained after repeated sub-culturing of isolated fungi. After the determination, the strains were preserved in liquid nitrogen in the form of freeze-dried conserves, and under paraffin oil on an agar slant in test tubes.

### 4.4. Determination of Microorganisms

A Phire Plant Direct PCR Kit was used to obtain DNA for preparation and PCR from five-day-old cultures grown on 2% malt extract agar. Translation elongation factor 1-alpha fragment strains belonging to genus *Fusarium* were amplified with primers EF1 and EF2 under conditions according to O’Donnell et al. [39]. Beta-tubulin fragment was amplified with primers Ben2f [40] and Bt2b [41] in strains belong to genera *Aspergillus* and *Penicillium*. PCR amplification of the section was performed in touch-down PCR mode: denaturation at 95 °C for 2 min; five cycles of 95 °C for 30 s, annealing for 30 s with temperature starting at 65 °C and decreasing by 1 °C each cycle and extension for 30 s; followed by 30 cycles of 95 °C for 15 s, 60 °C for 30 s, and 72 °C for 30 s; and final extension at 72 °C for 1 min. The obtained sequences were checked and compared using the Chromas and BioEdit programs. The species identity was determined by comparing the DNA sequence to the NCBI database using BLAST [42].

### 4.5. Antimicrobial Assay

The antifungal tests were carried out according to the method reported by Kloucek et al. [21] with several modifications. In the first phase of the study, all EOs were tested at the highest concentration (250 µL/L of air); then, 125 and 62.5 µL/L were tested. The tests were performed in 90 mm Petri dishes (PDs) divided into four sections. Into each section, 5 mL of SDA medium was poured, and SDA medium was poured onto the lid as well. After solidification, different mycelia were inoculated onto the middle of the three compartments by sterile loops. The fourth compartment remained empty as a purity control. The EOs were diluted in ethyl acetate to obtain the final volumes required. Each solution was equally distributed on 85 mm round sterile filter paper using a micropipette, and the paper was left to dry for 1 min for the evaporation of ethyl acetate. Finally, the filter paper was laid onto the walls of the compartments, so there was no direct contact with the medium in the Petri dish or lid containing solidified medium; then, the PD was hermetically closed and sealed with parafilm. The PDs were incubated at 25 °C in reverse position for 17 days, and every day the fungal growth was evaluated by measuring two perpendicular diameters of the colony using a ruler. The radial growth inhibition was observed and compared with blank filter paper with and without ethyl acetate, which served as negative controls. All the assays were carried out in triplicate and under aseptic conditions.

### 4.6. Statistical Analysis

The data were homogeneous and normal according to Bartlett and Shapiro tests. The results were tested by one-way ANOVA and Scheffe΄s method of homogeneous subsets (Statistica12, StatSoft CR s.r.o., Prague, Czech Rep.). The data were normalized to a percentage of fungal growth each day, with the growth of the control that day equaling 100%. A table containing the statistical data is available in the Appendix A (Appendix A).

## 5. Conclusions

In this study, we demonstrated that antifungal effects of the EOs from thyme, oregano, clove, and lemongrass can be achieved with lower doses and for longer time than those tested in previous studies. Complete inhibition was achieved at concentrations equal to or higher than 125 µL/L; at 62.5 µL/L, the fungal growth was significantly slowed down. The above discussion supports the suitability of the selected method, as studies have demonstrated a more efficient action of EOs in the gaseous phase. With this method, it is also possible to reduce the required amount of applied EO. However, this depends on the combination of the type of EO and the fungal strain. In general, the EOs tested can be ranked according to effectiveness as follows: thyme > oregano > lemongrass > clove. The most suitable uses would be EOs that have the broadest antifungal effect due to the real contamination of grains being by several strains. Future research needs to focus on testing these low-dose EOs in agricultural and food matrices, as they could become a part of organic production, could extend shelf life, and would not compromise sensory quality. In addition, our method is suitable for the screening of large quantities of samples in a shorter time and could be standardized for testing antimicrobial activity in the gas phase due to the uniformity of the headspace.

## Figures and Tables

**Figure 1 molecules-26-01313-f001:**
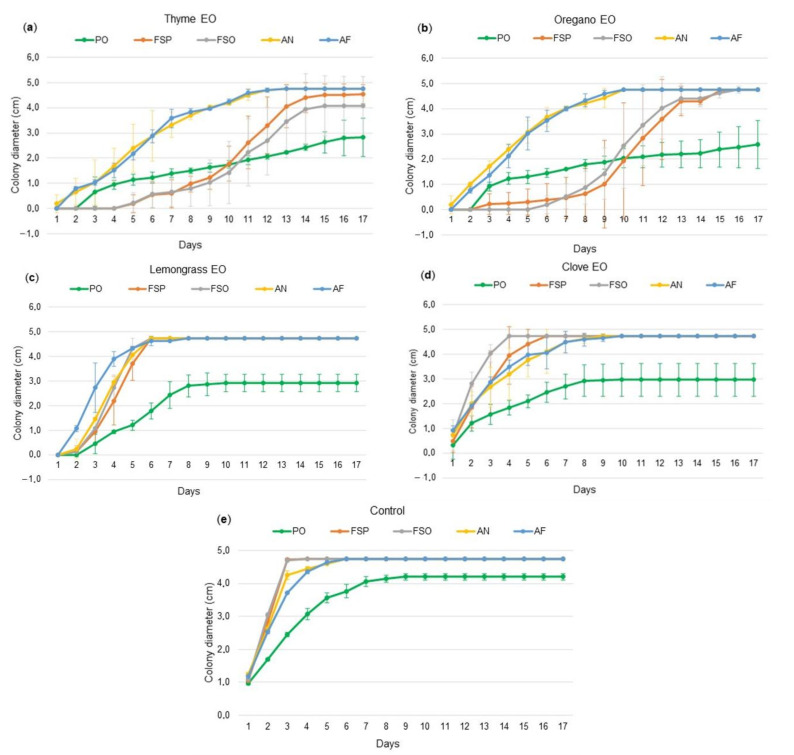
Each graph shows the efficiency of one essential oil (EO) at 62.5 µL/L against particular fungal strains (PO = *Penicillium ochrochloron*, FSP = *Fusarium sporotrichioides*, FSO = *F. solani*, AN = *Aspegillus niger*, and AF = *A. flavus*) over 17 days. The data are the average of three repetitions. (**a**) Efficiency of thyme EO; (**b**) efficiency of oregano EO; (**c**) efficiency of lemongrass EO; (**d**) efficiency of clove EO; (**e**) control sample without any EO.

## Data Availability

The data presented in this study are available in Figure 1 and Appendix A.

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
