# Peer review of "Inhibition of Fungal Strains Isolated from Cereal Grains via Vapor Phase of Essential Oils"

_molecules, 2021, doi:10.3390/molecules26051313_

Round 1
Reviewer 1 Report
The authors StĹ™elková et al. of the manuscript “Inhibition of fungal strains isolated from cereal grains by vapour phase of essential oils” underlined the antifungal properties of many essential oils. This seems of a great importance since the experiment is well conducted and the manuscript is written following a logical order. A few minor changes/clarifications and English revision are required before publishing:
Introduction
I suggest shortening the introduction, some paragraphs are too long (e.g. Today synthetic pesticides play a major role in crop protection….) and many details are written.
Please review this section and mention the information that are really needed there.
Line 35: Please add “European Food Safety Authority” not only the abbreviation “EFSA”
Results
Line 118: Please write “Aspergillus niger” as “A. niger”
Line 120: What the author trying to say in the following sentence “for the first time”
Line 139: “against which it was the least effective lemongrass EO” this an unclear statement
Line 140: Please correct this sentence “statistically significantly differ from the control”
Line 144: Change Penicillium ochrochloron to P. ochrochloron.
Line 145: Comma after “From all tested strains”
Discussion
Line 200: Please add comma after “of Mentha piperita”
Line 213: Remove “section” from the following sentence “Aspergillus section Flavi”
Line 218: Change Penicillium hrochloron to P. ochrochloron
Tables and figures
The authors mentioned about the effectiveness of the studied EOs at different concentrations in the text body but tables and figures demonstrate data without any information regarding the concentration in the titles of: Table 1: Fungal growth inhibition by EOs, Table S2: Heat map of EOs efficiency against fungal strains and Figure 1.
Author Response
Introduction: Thank you for your suggestion for shortening of this section, some inessential sentences have been removed, see changes in the manuscript.
Line 35: The “EFSA” has been replaced by “European Food Safety Authority”.
Line 118: Aspergillus niger has been changed to A. niger.
Line 120: The sentence “On the 6th and 7th day, for the first time, oregano treatment in both Aspergilli did not statistically differ from clove and lemongrass, respectively.” means that until 6th and 7th day the oregano treatment did statistically differ from clove and lemongrass and this timepoint is the first moment from which it did not differ.
Line 139: In the sentence “Clove EO (Fig. 1d) at this concentration was the least effective EO against all tested strains except A. flavus, against which was the least effective lemongrass EO.” has been changed “against which it was” to “against which was”. The statement means that against A. flavus was the least effective lemongrass EO, while against other fungi was least effective clove EO.
Line 140: The sentence “After 24 hours, only clove treatment did not statistically significantly differ from the control.” has been changed to “After 24 hours, only the clove treatment was statistically equal to the growth control.”.
Line 144: Penicillium ochrochloron has been changed to P. ochrochloron.
Line 145: Comma has been placed after “From all tested strains”.
Line 200: Comma has been placed after “of Mentha piperita”.
Line 213: The “section” has been removed from “Aspergillus section Flavi”.
Line 218: Penicillium ochrochloron has been changed to P. ochrochloron.
Tables and figures: Information about the concentration of tested EO has been added to the titles and descriptions of Table S1, Table S2 and Figure 1.
Reviewer 2 Report
This study evaluated the effect of vapor phase essential oils (EOs) on fungal strains isolated from cereal grains. This is an interesting and important area of research.
Specific Comments
- The authors evaluated the composition of EOs by GC-MS analysis, but almost not present these data for discussion. Please, add the literature data on the EO composition of the EOs studied.
- The composition of cajeput EO was not analyzed. Please explain this.
- The concentrations of EOs are indicated at the time of EO application. Is it possible to evaluate EO concentration in the dishes during the 17-day incubation period? It seems likely that the EO will be almost fully evaporated during the first 3-5 days. This could be evaluated by using a 1-3 day incubation with EO and then removing the filter papers with EOs from the Petri dishes to compare efficacy
- Would the presence of grain alter the MIC of EO treatment on the fungi?
- The authors indicate in Lines 13 and 77 that EOs are nontoxic. This is not always true and should be revised.
- Please review the manuscript for grammar.
Author Response
- A brief mentionof the chemical composition can be found in the discussion on the line 241, however, additional information has been added following the recommendation.
- Thank you for your comment, itwas not mentioned in the main text. Composition of cajeput EO has been added to the chapter 2.1.
- Comment:The concentrations of EOs are indicated at the time of EO application. Is it possible to evaluate EO concentration in the dishes during the 17-day incubation period? It seems likely that the EO will be almost fully evaporated during the first 3-5 days.  This could be evaluated by using a 1-3 day incubation with EO and then removing the filter papers with EOs from the Petri dishes to compare efficacy.
Answer: This comparison could be made, but it is assumed that not all of the EO leaked out, as some EOs inhibited some fungi for more than 3-5 days (e.g. thyme or oregano EO – Figure 1). In our design, the evaporation of the EOs from the Petri dish was minimized by the agar in the Petri dish lid and sealing by the parafilm. Another factor could be the different volatility of the individual components of the EOs or their possible transformation by the fungi themselves. In addition, a certain amount is absorbed into the agar and thus remains in the inner space, as found by Inouye at al. (2000).
Inouye, S.; Tsuruoka, T.; Watanabe, M.; Takeo, K.; Akao, M.; Nishiyama, Y.; Yamaguchi, H. Inhibitory effect of essential oils on apical growth of Aspergillus fumigatus by vapour contact. Mycoses 2000, 43, 17–23, doi:10.1046/j.1439-0507.2000.00538.x.
- Comment: Would the presence of grain alter the MIC of EO treatment on the fungi?
Answer: The presence of grains may alter the MIC of EO treatment, but not significantly. Our team has already done a similar study with oat grains (number 13 in the list of references), and by slightly higher concentrations significant inhibition was achieved. However, this was not the intension of this study.
- In line 13 and 79 has been revised the statement about nontoxicity of EOs.
- The manuscript has been reviewed for grammar, thank you for your suggestion.
Reviewer 3 Report
Manuscript The inhibition of fungal strains isolated from cereal grains by the gas phase of the essential oils is really attractive. It is well written and therefore easy to read. It also has a lot of scientific and application significance. As you mentioned in the introduction, the problem of mold growth in grains and grain products causes food spoilage. Moreover, molds are a source of mycotoxins, which is a serious threat to human life and health. I have few little comments.
"However, even at concentrations that caused smaller than 100% mycelial growth inhibition, which is also our case, conidia have lost their pigmentation (became hyaline). According to Yigit et al. [33] this effect might
decrease virulence of the pathogen." Can you explain it?
In the text you write down "In addition, our results suggest that, thanks to the use of the gas phase, it is possible to achieve inhibition of fungal growth using significantly lower concentrations than have been tested so far...Higher concentrations could adversely affect sensory properties and would require more EOs to be used, which would be reflected in costs..." You suggest that oils (EOs) in the vapour phase give a chance to apply lower concentration of it and thanks to that - it would not affect sensory properties of cereals and cereal's products. Generally I am agree with you but it should be confirmed by sensory panel. Of course I know it may be difficult at this moment and I do not required that - it is only a suggestion for the future.
Maybe it is worth commenting on how the use of the EOs influenced the growth of mold biochemically, even purely hypothetically. Sometimes it is worth making a hypothesis in the discussion section.
Figure 1 is not readable.
In chapter 4.3. it is worth mentioning how the molds were stored - there are different methods: freezing, filled with sterile paraffin, etc.
"The species identity was determined by comparing the DNA sequence to the NCBI database using BLAST." For the on-line BLAST web interface provided by NCBI, and if the context allows it, you could cite the "The BLAST Sequence Analysis Tool" chapter in the NCBI Handbook:
Madden T. The BLAST Sequence Analysis Tool. 2002 Oct 9 [Updated 2003 Aug 13]. In: McEntyre J, Ostell J, editors. The NCBI Handbook [Internet]. Bethesda (MD): National Center for Biotechnology Information (US); 2002-. Chapter 16. Available from: http://www.ncbi.nlm.nih.gov/books/NBK21097/
If this is not an option, there are a number of other options, including:
NCBI Resource Coordinators (2013) "Database resources of the National Center for Biotechnology Information"
Boratyn, G.M. et al. (2013) "BLAST: a more efficient report with usability improvements"
Johnson M. et al. (2008) "NCBI BLAST: a better web interface"
Chromas and BioEdit programs aslo have to be cited, if possible.
Author Response
- Comment:"However, even at concentrations that caused smaller than 100% mycelial growth inhibition, which is also our case, conidia have lost their pigmentation (became hyaline). According to Yigit et al. [33] this effect might decrease virulence of the pathogen." Can you explain it?
Answer: Yigit et al. (2000) reported that the mycelium after exposure to EO showed a low density of conidia and conidiophores and that the colour of the mycelium was pale. This effect might decrease virulence of the pathogen, so that incidence of disease would decrease. Also, Gastebois et al. (2009) states that increased fungal virulence is associated with increased synthesis of melanin. But the main idea was that even subinhibitory concentrations could inhibit some fungal activity.
Gastebois, A.; Clavaud, C.; Aimanianda, V.; Latgé, J.P. Aspergillus fumigatus: Cell wall polysaccharides, their biosynthesis and organization. Future Microbiol. 2009, 4, 583–595, doi:10.2217/fmb.09.29.
- Comment: You suggest that oils (EOs) in the vapour phase give a chance to apply lower concentration of it and thanks to that – it would not affect sensory properties of cereals and cereal's products. Generally, I agree with you but it should be confirmed by sensory panel. Of course, I know it may be difficult at this moment and I do not require that - it is only a suggestion for the future.
Answer: Yes, of course we agree with you, but at this stage of research it was not part of this experiment. It is possible to look at another article of our group under the number 13 in the list of references, which also includes sensory evaluation. However, we also plan to perform sensory evaluation in the next phase.
- Comment: Maybe it is worth commenting on how the use of the EOs influenced the growth of mold biochemically, even purely hypothetically. Sometimes it is worth making a hypothesis in the discussion section.
Answer: In the introduction (line 83-86) we mentioned purely hypothetically how EOs influenced the growth of fungi: „It is postulated that EOs, through their lipophilicity, have the ability to penetrate the plasma membrane, causing morphological changes in the hyphae, damaging the enzymatic cell systems, disrupting in the plasma membrane, and eventually destroying the mitochondria, thereby killing the fungi.” We did not discuss it with other authors, because we did not perform any experiment focused on the mechanism of action.
- We apologize, but after opening the document, event after printing, Figure 1 seems readable to us. We believe that the editors would require improvement for the final version, if it is needed.
- At the end of chapter 4.3 information has been added about how the fungi have been stored after their identification.
- In the chapter 4.4 has been cited the "The BLAST Sequence Analysis Tool" chapter in the NCBI Handbook. It could be found under the number 42.
- The manuscript has been reviewed for grammar.
Round 2
Reviewer 2 Report
The revised manuscript addressed my concerns.
Reviewer 3 Report
All corrections have been made, thank you also for answering my questions sufficiently.